# A peridomestic *Aedes malayensis* population in Singapore can transmit yellow fever virus

Elliott F. Miot[1,2,3]*, Fabien Aubry[1], Stéphanie Dabo[1], Ian H. Mendenhall[4], Sébastien Marcombe[3], Cheong H. Tan[5], Lee C. Ng[5], Anna-Bella Failloux[6], Julien Pompon[4,7], Paul T. Brey[3☯], Louis Lambrechts[1☯]*

**1** Insect-Virus Interactions Unit, Institut Pasteur, UMR2000, CNRS, Paris, France, **2** Sorbonne Université, Collège doctoral, Paris, France, **3** Medical Entomology and Vector-Borne Disease Unit, Institut Pasteur du Laos, Vientiane, Lao PDR, **4** Program in Emerging Infectious Disease, Duke-NUS Medical School, Singapore, **5** Environmental Health Institute, National Environment Agency, Singapore, **6** Arboviruses and Insect Vectors Unit, Institut Pasteur, Paris, France, **7** MIVEGEC, IRD, CNRS, Univ. Montpellier, Montpellier, France

☯ These authors contributed equally to this work.
* elliott.miot@gmail.com (EFM); louis.lambrechts@pasteur.fr (LL)

**Data Availability Statement:** All relevant data are within the manuscript and its Supporting Information files.

## Abstract

The case-fatality rate of yellow fever virus (YFV) is one of the highest among arthropod-borne viruses (arboviruses). Although historically, the Asia-Pacific region has remained free of YFV, the risk of introduction has never been higher due to the increasing influx of people from endemic regions and the recent outbreaks in Africa and South America. Singapore is a global hub for trade and tourism and therefore at high risk for YFV introduction. Effective control of the main domestic mosquito vector *Aedes aegypti* in Singapore has failed to prevent re-emergence of dengue, chikungunya and Zika viruses in the last two decades, raising suspicions that peridomestic mosquito species untargeted by domestic vector control measures may contribute to arbovirus transmission. Here, we provide empirical evidence that the peridomestic mosquito *Aedes malayensis* found in Singapore can transmit YFV. Our laboratory mosquito colony recently derived from wild *Ae. malayensis* in Singapore was experimentally competent for YFV to a similar level as *Ae. aegypti* controls. In addition, we captured *Ae. malayensis* females in one human-baited trap during three days of collection, providing preliminary evidence that host-vector contact may occur in field conditions. Finally, we detected *Ae. malayensis* eggs in traps deployed in high-rise building areas of Singapore. We conclude that *Ae. malayensis* is a competent vector of YFV and re-emphasize that vector control methods should be extended to target peridomestic vector species.

## Author summary

Yellow fever is a dreadful disease caused by a mosquito-borne virus circulating in Africa and South America. Historically, the Asia-Pacific region has remained free of yellow fever but the ever increasing influx of travelers puts places such as Singapore at unprecedented risk of yellow fever virus introduction. The present study characterized the potential contribution of a mosquito species called *Aedes malayensis* to yellow fever virus transmission

**Funding:** EFM was supported by a Calmette and Yersin doctoral fellowship of the Institut Pasteur International Network. This work was funded by Agence Nationale de la Recherche (grant ANR-17-ERC2-0016-01 to LL), the French Government's Investissement d'Avenir program Laboratoire d'Excellence Integrative Biology of Emerging Infectious Diseases (grant ANR-10-LABX-62-IBEID to LL), the European Union's Horizon 2020 research and innovation program under ZikaPLAN grant agreement no. 734584 (to LL), and the City of Paris Emergence(s) program in Biomedical Research (to LL). The funders had no role in study design, data collection and analysis, decision to publish, or preparation of the manuscript.

**Competing interests:** The authors have declared that no competing interests exist.

in Singapore. *Aedes malayensis* breeds in urban parks of Singapore and is suspected to have participated in the resurgence of other mosquito-borne diseases such as dengue because it is not targeted by current mosquito control measures. Not only was *Ae. malayensis* able to experimentally acquire and transmit yellow fever virus, but it was also found to engage in contact with humans in a field situation. This empirical evidence indicates that *Ae. malayensis* is a competent vector of yellow fever virus and should be targeted by mosquito control programs.

## Introduction

The case-fatality rate of yellow fever virus (YFV) ranges from 15% to 50% [1] and is one of the highest among arthropod-borne viruses (arboviruses). YFV is endemic in 47 countries in Africa and South America, with an estimated annual incidence of around 200,000 cases and 30,000 deaths [2]. Despite an efficient vaccine against YFV, the past few years have seen a growing number of YFV outbreaks (Democratic Republic of the Congo, Angola, Uganda, Brazil and most recently Nigeria) [3–5]. During such outbreaks, increasing numbers of unvaccinated travelers who become infected and return to non-endemic countries have raised the risk of YFV introduction to unprecedented levels [6]. The Asia-Pacific region has remained free of YFV until now but the risk of introduction has never been higher [6–8]. In 2016, eleven Chinese workers infected with YFV in Angola who returned to China were the first cases of YFV diagnosed in travelers to Asia [7]. Over two billion immunologically naïve people live in Asia and the current vaccine production capacity would be insufficient to prevent a massive YFV epidemic, while mosquito control programs would be overwhelmed [8].

With 18.5 million visitors in 2018, Singapore is a global hub for tourism, trade and transport that includes one of the busiest ports in the world and a major international airport. These features put Singapore at high risk for introduction of arboviruses, including YFV. Strict and sustained vector control measures together with household structural improvements, currently achieve very low densities of the domestic arbovirus vector *Aedes aegypti* [9]. Re-emergence of dengue virus (DENV), chikungunya virus (CHIKV) and Zika virus in Singapore in the last two decades [10–13] supports the hypothesis that peridomestic mosquito species untargeted by vector control measures may contribute to 'cryptic' arbovirus transmission [14]. In particular, the peridomestic mosquito *Aedes malayensis* (a member of the *Stegomyia* subgenus) breeds in urban parks of Singapore and is experimentally competent for DENV and CHIKV [14].

Here, we evaluated the potential contribution of the peridomestic mosquito *Ae. malayensis* to YFV transmission in Singapore. Vector competence assays in the laboratory and a small-scale field survey provided evidence that indeed *Ae. malayensis* could contribute to YFV transmission in Singapore.

## Methods

### Ethics statement

This study used human blood samples to prepare mosquito artificial infectious blood meals. Healthy donor recruitment was organized by the local investigator assessment using medical history, laboratory results and clinical examinations. Biological samples were supplied through participation of healthy volunteers at the ICAReB biobanking platform (BB-0033-00062/ICAReB platform/Institut Pasteur, Paris/BBMRI AO203/[BIORESOURCE]) of the Institut

Pasteur to the CoSImmGen and Diagmicoll protocols, which have been approved by the French Ethical Committee Ile-de-France I. The Diagmicoll protocol was declared to the French Research Ministry under reference DC 2008–68 COL 1. The use of human-baited double net trap was approved by the National Environment Agency (NEA) of Singapore (NEA/PH/CLB/19-00004). All adult subjects provided written informed consent.

## Mosquitoes

Experiments were carried out with a laboratory colony derived in 2014 from a wild population of *Ae. malayensis* in Singapore and subsequently maintained at the Duke-NUS Medical School for >50 generations [14]. The 8th generation of a laboratory colony of *Ae. aegypti* maintained at the Institut Pasteur in Laos was used as a control. The *Ae. malayensis* colony was initiated with mosquito eggs collected from two forested areas located in the Northern (Sembawang) and the Southern (East Coast Park) regions of Singapore [14]. The *Ae. aegypti* colony originated in the town of Paksan, Paksan district, Bolikhamsai province, Laos. Mosquitoes were reared under controlled insectary conditions (28˚C, 70% relative humidity, 12:12 hour light cycle). Eggs were hatched synchronously in a vacuum chamber for 1 hour. Larvae were reared in 24 x 34 x 9 cm plastic trays containing 1.5 L of dechlorinated tap water and supplemented with Tetramin (Tetra) fish food at a density of 400 larvae per tray. Six hundred adults were kept in 30 x 30 x 30 cm Bugdorm-1 insect cages with permanent access to 10% sucrose solution.

## Virus

Vector competence experiments were carried out with a low-passage YFV isolate (YFV-S79 strain) belonging to the West African lineage, which was originally obtained in 1979 from the serum of a patient returning to France from Senegal [15]. Prior to its use in the experiments, the YFV isolate was passaged twice in newborn mouse brains and three times in *Aedes albopictus* C6/36 cells. Virus stock was produced during the last passage, as previously described for DENV [16]. Virus titration was performed by standard focus-forming assay (FFA), as previously described for DENV [16] with the exception of the primary antibody. A mouse anti-flavivirus group antigen monoclonal antibody MAB10216 (Merck Millipore) diluted 1:1,000 in phosphate-buffered saline supplemented with 1% bovine serum albumin (Interchim) was used as primary antibody.

## Virus challenge

In two separate experiments (referred to as experiments 1 and 2 hereafter), mosquitoes were orally challenged with YFV as previously described for DENV [16]. Briefly, eight-day-old females deprived of sucrose for 24 hours were offered an artificial infectious blood meal in 3 rounds of 15 min using a Hemotek membrane feeding apparatus with porcine intestine as the membrane. Blood meals consisted of a 2:1 mix of washed human erythrocytes and virus suspension. Adenosine triphosphate (Merck) was added as a phagostimulant [17] to the blood meal at a final concentration of 10 mM. To obtain a dose-response curve, three separate blood meals with increasing YFV concentrations were prepared. In experiment 1, mosquitoes were exposed to 2.9 x $10^6$ focus-forming units (FFUs)/mL, 2.1 x $10^5$ FFUs/mL, or 1.8 x $10^4$ FFUs/mL of YFV. In experiment 2, they were exposed to 8.1 x $10^5$ FFUs/mL, 5.4 x $10^4$ FFUs/mL, or 3.6 x $10^3$ FFUs/mL of YFV. Fully engorged females were sorted on wet ice, transferred into 0.5-L cardboard containers and maintained in a climatic chamber (28˚C, 70% relative humidity, 12:12 hour light cycle) with permanent access to 10% sucrose solution. At 10 and 14 days post blood meal, mosquitoes were cold anesthetized (experiment 1) or paralyzed with

triethylamine (experiment 2) to perform *in vitro* salivation [18]. In experiment 1, the wings and legs were removed and stored at -80°C prior to the salivation assay. In experiment 2, the heads were removed and stored at -80°C after the salivation assay. The proboscis of each female was inserted into a 20-μL pipet tip containing 5 μL of fetal bovine serum (FBS). After 30 min of salivation, the body remainders were stored dry at -80°C. The saliva-containing FBS was mixed with 45 μL of Leibovitz's L-15 medium and immediately inoculated onto C6/36 cells for titration by FFA, as described above, without subsequent dilution.

## Virus detection

Bodies were homogenized individually in 300 μL of lysis buffer (Tris 10 mM, NaCl 50 mM, EDTA 1.27 mM with a final pH adjusted to 8.2) supplemented with proteinase K (1 μL for 55.5 μL of buffer). Body homogenates were centrifuged and 100 μL of supernatant was incubated for 5 min at 56°C then 10 min at 98°C to extract viral RNA. Detection of YFV RNA was performed using a two-step RT-PCR reaction to generate a 192-bp amplicon located in a conserved region of the *NS3* gene of YFV. Total RNA was reverse transcribed into cDNA with random hexamers using M-MLV reverse transcriptase (ThermoFisher Scientific) using the following program: 10 min at 25°C, 50 min at 37°C and 15 min at 70°C. The cDNA was subsequently amplified using DreamTaq DNA polymerase (ThermoFisher Scientific). The 20-μL reaction volume contained 1x of reaction mix and 10 μM of primers (forward: 5'-GCGTAAG GCTGGAAAGAGTG-3'; reverse: 5'-CTTCCTCCCTTCATCCACAA-3'; [19]). The thermocycling program was 2 min at 95°C, 35 cycles of 30 sec at 95°C, 30 sec at 60°C, and 30 sec at 72°C with a final extension step of 7 min at 72°C. Amplicons were visualized by electrophoresis on a 2% agarose gel. Mosquito wings/legs and heads were homogenized in 300 μL of Leibovitz's L-15 medium supplemented with 2% FBS, centrifuged, and titrated by FFA as described above.

## Host-vector contact

Adult mosquito sampling was conducted in a forested area in the northern part of Singapore (Sembawang; 1.456493, 103.825983) in March 2019. Host-vector contact was evaluated using a human-baited double net trap as previously described [20]. Briefly, a human volunteer was resting on a long chair inside a small, untreated bed-net protected from mosquitoes during daylight hours (7 a.m. to 7 p.m.). A larger bed-net covered the smaller one with a 30-cm opening at the bottom allowing mosquitoes to enter the 20-cm space between bed-nets. A large plastic sheet was placed above the bed-nets as a protection. Every 10 min, the human volunteer collected mosquitoes trapped between bed-nets using a mouth aspirator. Aspirated mosquitoes were transferred into holding cups, killed and grouped per 1-hour interval. Back in the laboratory, mosquitoes were morphologically identified and stored at -80°C.

## Gravitrap survey

The presence of *Ae. malayensis* in Singapore's high-rise public housing was assessed using surveillance data obtained in 2018 from the Singapore NEA. The island-wide Gravitrap surveillance network was implemented by NEA to monitor the spatial and temporal variability of adult *Aedes* mosquito populations and to evaluate the effectiveness of vector control measures. In 2018, approximately 53,000 Gravitraps were deployed in about 9,200 public apartment blocks across the island. The Gravitrap is a simple, hay infusion-filled cylindrical trap with a sticky inner surface to catch ovipositing gravid female *Aedes* mosquitoes [21].

## Statistical analyses

Vector competence was estimated by measuring virus infection, dissemination and transmission rates. Infection rate was estimated as the proportion of YFV-positive mosquitoes among all mosquitoes tested. Dissemination rate was estimated as the proportion of infected mosquitoes with YFV-positive wings and legs (experiment 1) or head (experiment 2). Transmission rate was estimated as the proportion of mosquitoes with a disseminated infection that had YFV-positive saliva. In addition, transmission efficiency was calculated as the proportion of mosquitoes with infectious saliva among all mosquitoes tested. Vector competence indices were analyzed with a logistic regression model in which each individual mosquito was associated with a binary variable (1 = YFV-positive or 0 = YFV-negative), followed by an analysis of deviance with the R package *car* [22]. An initial analysis showed that infection rates were not statistically significantly different between 10 and 14 days post blood meal and both time points were subsequently combined. The analysis of dissemination and transmission rates was restricted to 14 days post blood meal because the two time points differed and there was too little data on day 10 for a meaningful analysis. The experiment effect was statistically non-significant overall and removed from the model. For infection and dissemination rates, the final regression model included the infectious dose ($\log_{10}$-transformed blood meal titer), the mosquito species (*Ae. aegypti* and *Ae. malayensis*), and their interaction. The 50% oral infectious dose ($OID_{50}$) and 10% oral infectious dose ($OID_{10}$) values and their respective 95% confidence intervals were derived from the logistic fits of infection rates using the R package *MASS* [23]. The analysis of transmission rate was restricted to the highest infectious dose and the final regression model only included the mosquito species. All statistical analyses were performed using the R software version 3.5.2 [24] and graphical representations were generated with the R packages *ggplot2* [25] and *ggpubr* [26].

## Results

We measured the vector competence of 68 and 31 *Ae. malayensis* females and 132 and 76 *Ae. aegypti* females in experiments 1 and 2, respectively. The data from both experiments were combined because initial analyses showed that none of the vector competence indices differed significantly between them. In each experiment, mosquitoes were exposed to three different YFV infectious doses to establish a dose-response curve. Vector competence was analyzed 14 days post blood meal, with the exception of infection rates that also included mosquitoes sampled at 10 days post blood meal.

The proportion of blood-fed females that became infected (i.e., the infection rate) increased with the infectious dose (Fig 1). The highest infectious dose ($2.9 \times 10^{6}$ FFUs/mL) resulted in 74.5% and 75.6% infected females for *Ae. malayensis* and *Ae. aegypti*, respectively. Infectious dose was the only statistically significant ($p < 0.0001$) predictor of infection rate (Table 1), although there was a slight difference in the shape of the dose-response curve between the two species (Fig 1). Although the $OID_{50}$ was similar between the two species, the $OID_{10}$ was lower for *Ae. aegypti*. The $OID_{50}$ was 5.97 $\log_{10}$ FFUs/mL (95% CI = 5.81–6.14 $\log_{10}$ FFUs/mL) for *Ae. aegypti* and 6.13 $\log_{10}$ FFUs/mL (95% CI = 6.02–6.25 $\log_{10}$ FFUs/mL) for *Ae. malayensis*. The $OID_{10}$ was 5.03 $\log_{10}$ FFUs/mL (95% CI = 4.78–5.28 $\log_{10}$ FFUs/mL) for *Ae. aegypti* and 5.59 $\log_{10}$ FFUs/mL (95% CI = 5.41–5.78 $\log_{10}$ FFUs/mL) for *Ae. malayensis*.

The proportion of infected females that developed a disseminated infection (i.e., the dissemination rate) also increased with infectious dose (Fig 1). The highest infectious dose resulted in 42.9% and 57.1% dissemination in *Ae. malayensis* and *Ae. aegypti*, respectively. Infectious dose was the only statistically significant ($p < 0.0001$) predictor of dissemination

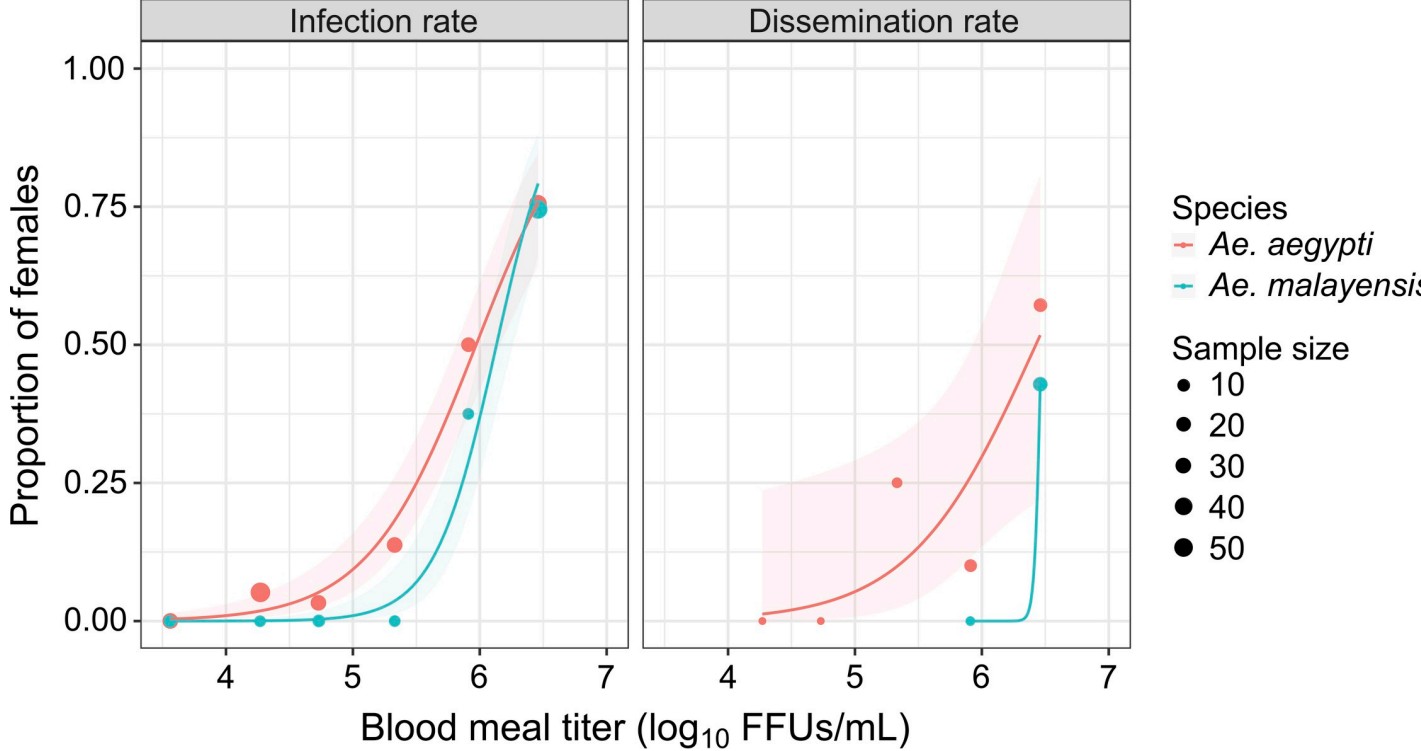

**Fig 1. Peridomestic *Ae. malayensis* in Singapore are orally susceptible to YFV.** Dose-response curves are shown for the *Ae. malayensis* population from Singapore under study and a control *Ae. aegypti* population. Infection rate is the proportion of blood-fed females testing YFV-positive 10–14 days post blood meal. Dissemination rate is the proportion of YFV-infected females with YFV-positive wings/legs or head 14 days post blood meal. The line represents the logistic regression of the data and the shaded area is the 95% confidence interval of the fit. The data shown are combined from two separate experiments.

rate (Table 1). Together, these results indicated that although *Ae. aegypti* was slightly more susceptible than *Ae. malayensis* at low infectious doses, the two species had a similar YFV susceptibility overall.

The proportion of females with a disseminated infection that had YFV-positive saliva (i.e., the transmission rate) was measured at 14 days post blood meal only for the highest infectious dose (Fig 2). YFV was detected in the saliva of 33.3% (3/9) of *Ae. malayensis* females and 37.5% (3/8) of *Ae. aegypti* females, and the difference between the two species was not statistically significant ($p = 0.8576$). Vector competence can be summarized by the overall proportion of blood-fed females that had YFV-positive saliva at 14 days post blood meal at the highest infectious dose (i.e., the transmission efficiency). Transmission efficiency was 11.5% (3/26) for *Ae. malayensis* and 14.3% (3/21) for *Ae. aegypti* (Fig 2), and the difference between the two species was not statistically significant ($p = 0.7795$).

**Table 1. Test statistics of YFV infection and dissemination rates.** Infection and dissemination rates were analyzed by logistic regression. The model included the effect of the YFV oral infectious dose ($\log_{10}$-transformed blood meal titer), the mosquito species (*Ae. aegypti* or *Ae. malayensis*) and their interaction.

| | Infection rate | | | Dissemination rate | | |
|---|---|---|---|---|---|---|
| | LR $\chi^2$ | Df | *p* value | LR $\chi^2$ | Df | *p* value |
| Dose | 97.42 | 1 | <0.0001 | 5.504 | 1 | 0.0190 |
| Species | 3.325 | 1 | 0.0682 | 1.040 | 1 | 0.3079 |
| Dose x Species | 3.099 | 1 | 0.0783 | 1.001 | 1 | 0.3171 |

Df: degrees of freedom; LR: likelihood ratio.

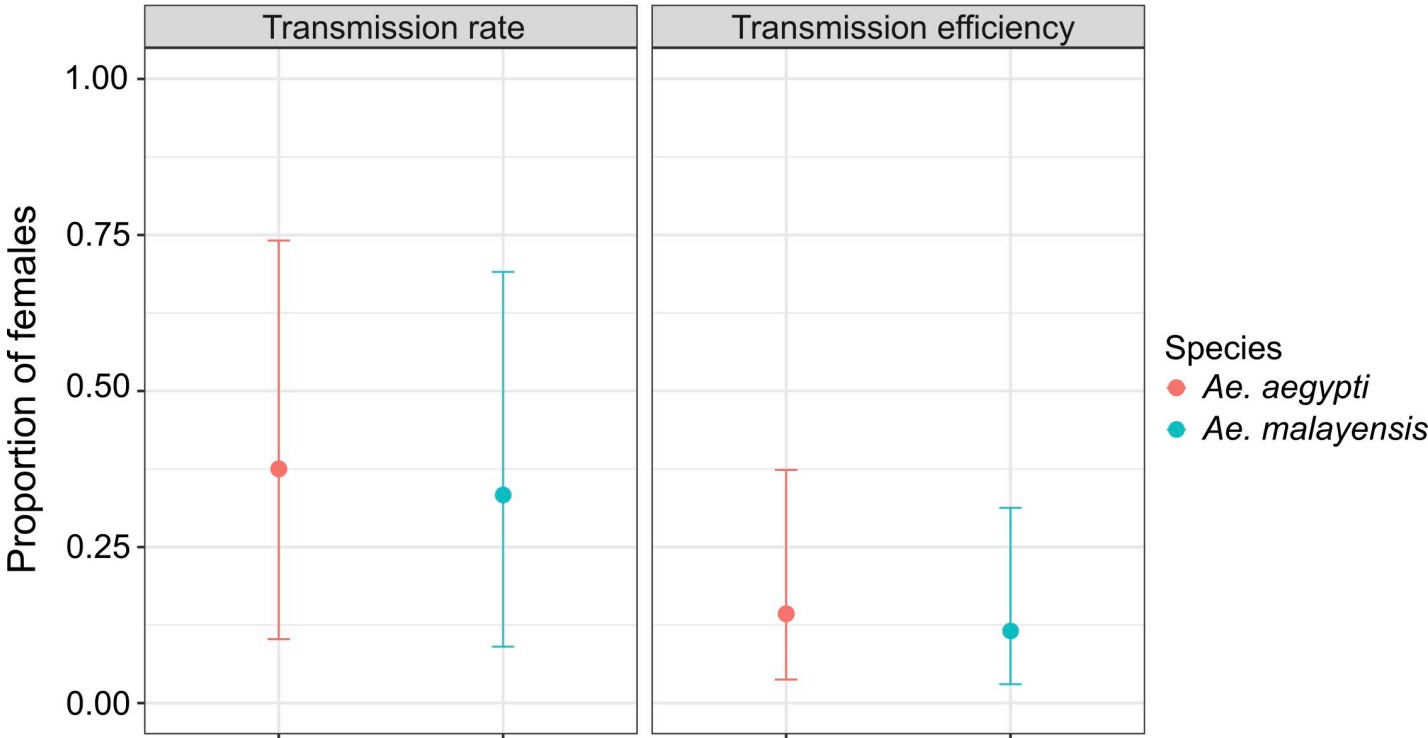

**Fig 2. Peridomestic *Ae. malayensis* in Singapore are competent for YFV transmission.** Transmission potential is shown for the *Ae. malayensis* population from Singapore under study and a control *Ae. aegypti* population 14 days after oral exposure to 2.9 x 10$^6$ FFU/mL of YFV. Transmission rate is the proportion of females with a disseminated infection that had infectious YFV in their saliva 14 days post blood meal. Transmission efficiency is the proportion of all blood-fed females that had infectious YFV in their saliva 14 days post blood meal. Vertical bars represent 95% confidence intervals of the proportions. The data shown are from two separate experiments.

In addition to vector competence, the ability of *Ae. malayensis* to transmit YFV depends on its propensity to bite humans. To assess host-vector contact of peridomestic *Ae. malayensis* in Singapore, we surveyed day-biting mosquitoes in a forested area of Sembawang, Singapore in

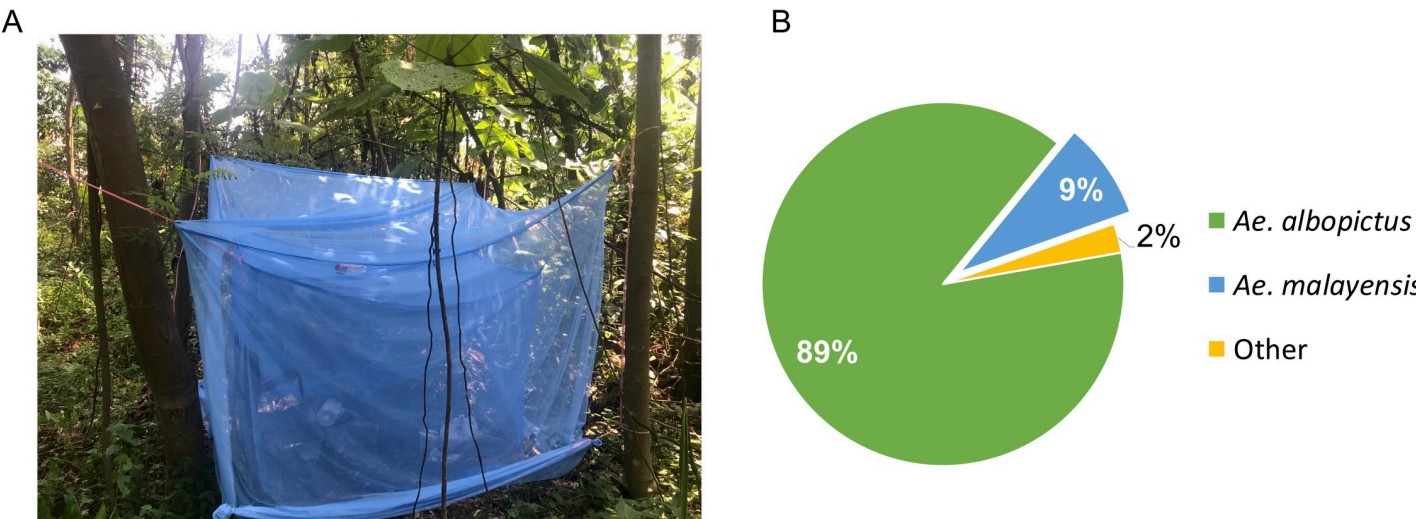

**Fig 3. Peridomestic *Ae. malayensis* in Singapore are attracted to humans.** (A) Human-baited double net trap setup in the forested area of Sembawang, Singapore. (B) Distribution of mosquito species among 115 female specimens collected at daytime in the human-baited double net trap.

March 2019 using a human-baited double net trap (Fig 3A). A total of 115 female mosquitoes were captured during three days of collection. The most abundant species was *Ae. albopictus* (89%) followed by *Ae. malayensis* (9%) (Fig 3B).

Out of 504,771 mosquitoes collected in Gravitraps deployed by NEA in Singapore's high-rise public apartment blocks in 2018, the predominant mosquito species was *Ae. aegypti* (80% of all captures). However, a total of 1,931 *Ae. malayensis* (0.38%) were caught in 1,741 traps. The average capture rate was 0.0013 *Ae. malayensis* per trap and per week, with an average of 1.1 specimen per *Ae. malayensis*-positive trap.

## Discussion

Our study indicates that peridomestic *Ae. malayensis* mosquitoes can contribute to YFV transmission in Singapore based on two lines of evidence. First, our laboratory mosquito colony recently derived from a wild *Ae. malayensis* population in Singapore was experimentally competent for YFV, to a level that was not statistically different from that of the *Ae. aegypti* controls. Adding to earlier data demonstrating vector competence of this colony for DENV and CHIKV [14], our results are the first experimental evidence, to our knowledge, that peridomestic *Ae. malayensis* from Singapore can potentially transmit YFV acquired from an infectious blood meal. Second, we captured *Ae. malayensis* females in a human-baited trap. Although our host-seeking experiment lacked a non-baited control trap, it provided preliminary evidence that this peridomestic *Ae. malayensis* population in Singapore may engage in human-biting behavior, as was already observed for other populations of the species in South-East Asia [27,28].

Despite its wide distribution in South-East Asia with records in Laos, Thailand, Cambodia, Vietnam, Peninsular Malaysia, the Andaman and Nicobar Islands, and Taiwan [27–30], the mosquito *Ae. malayensis* has been relatively poorly studied. Recently, however, *Ae. malayensis* was found prevalent in numerous urban parks of Singapore [14]. Not only were *Ae. malayensis* eggs collected in the forested areas of the parks but they were also found in some open-air areas frequented by humans [14]. The current study reports that in 2018, the National Vector Surveillance program in Singapore detected the presence of *Ae. malayensis* in Gravitraps [21] outside the urban parks of the island. Although the majority of *Aedes* mosquitoes found in public apartment blocks were *Ae. aegypti*, *Ae. malayensis* was detected in Gravitraps deployed in high-rise building areas, underlining the potential of this mosquito species to colonize urban habitats. It was recently hypothesized that peridomestic mosquito species, such as *Ae. malayensis*, which escape vector control measures targeting domestic vector species, may have contributed to the (re-)emergence of arboviruses in Singapore in the last two decades [14]. The well-known peridomestic mosquito *Ae. albopictus*, which was highly abundant in our human-baited trap survey, significantly increases the risk of arbovirus transmission in Singapore [14]. Control of peridomestic mosquitoes is a challenge because in addition to artificial breeding sites, less accessible natural sites such as tree holes need to be targeted as well [31].

Although we did not measure other parameters that influence the vectorial capacity of a mosquito species, such as adult density and survival, and the duration of the virus extrinsic incubation period [32], vector competence and attraction to humans make *Ae. malayensis* a potential vector of YFV and other arboviruses that deserves increased attention from public health authorities. The level of *Ae. malayensis* vector competence for YFV that we observed was in the same range as our *Ae. aegypti* controls from Laos, and as Brazilian *Ae. aegypti* that were orally challenged with the same virus isolate in a previous study [33]. It is worth noting, however, that vector competence results cannot be readily compared if they did not use the same oral infectious dose, or if other factors such as virus strain and detection assays were different. The dose-response curves that we generated in the present study will be useful for

future comparisons because they provide absolute estimates of vector competence indices, such as the $OID_{50}$. Moreover, two vector species or populations may have a similar level of vector competence at a given infectious dose, but vary at a different dose. For example, the DENV type 2 susceptibility of two field-derived populations of *Ae. aegypti* in Thailand were similar at a medium dose, but different at a low dose [34]. This example indicates that the minimal threshold for infection may differ between mosquito populations. Likewise, in the present study, we observed similar YFV $OID_{50}$ values for *Ae. aegypti* and *Ae. malayensis* but *Ae. aegypti* had a lower YFV $OID_{10}$ value. In natural conditions, the oral infectious dose is the primary driver of successful human-to-mosquito transmission of arboviruses [35,36].

Despite an efficient vaccine, YFV still remains an important public health concern in Africa and South America. YFV is maintained in sylvatic cycles between non-human primates and arboreal mosquitoes, and human outbreaks occur when the virus is introduced into an urban cycle mediated by *Ae. aegypti* [37]. Although there is no evidence of a sylvatic cycle of YFV in the Asia-Pacific region, the risk of spillover transmission exists. The lack of YFV outbreaks in the Asia-Pacific region to date could be due to cross-protective immunity conferred by other widespread flaviviruses such as DENV and Japanese encephalitis virus, but it could also reflect relatively fewer opportunities for YFV introduction until now. Rampant globalization now puts the Asia-Pacific region at an unprecedented risk of YFV introduction [6–8]. Billions of immunologically naïve people, together with competent *Ae. aegypti* populations [38,39], create a dangerous set of conditions that are theoretically favorable to a massive YFV epidemic. On the other hand, a scenario whereby YFV would establish a novel sylvatic cycle [40] seems unlikely in Singapore given the small size of resident non-human primate populations. Clearly, places with high densities of the domestic vector *Ae. aegypti* are most at risk of spillover transmission, but we emphasize that in the absence of *Ae. aegypti*, ancillary vectors should not be overlooked. Competent mosquito species such as *Ae. malayensis* could facilitate introduction of YFV into the Asia-Pacific region. In Singapore, strict and sustained vector control measures have achieved low densities of *Ae. aegypti* [9] but this has failed to prevent arbovirus outbreaks in the last two decades [10]. Following a previous study on the ability of *Ae. malayensis* from Singapore to transmit DENV and CHIKV [14], we reiterate the need to account for peridomestic species in the management of mosquito vector populations.

## Acknowledgments

We thank Catherine Lallemand for assistance with mosquito rearing and three anonymous reviewers for constructive comments on an earlier version of the manuscript. We are grateful to the blood donor volunteers for participation in this study and the ICAReB staff for its support. We thank the National Park Board Singapore for their permission to collect in their parks and Jeffrey C. Hertz for his support and advice with the mosquito collections.

## Author Contributions

**Conceptualization:** Elliott F. Miot, Paul T. Brey, Louis Lambrechts.

**Formal analysis:** Elliott F. Miot, Louis Lambrechts.

**Funding acquisition:** Paul T. Brey, Louis Lambrechts.

**Investigation:** Elliott F. Miot, Fabien Aubry, Stéphanie Dabo, Cheong H. Tan, Lee C. Ng.

**Resources:** Ian H. Mendenhall, Sébastien Marcombe, Anna-Bella Failloux, Julien Pompon, Louis Lambrechts.

**Supervision:** Paul T. Brey, Louis Lambrechts.

**Visualization:** Elliott F. Miot.

**Writing – original draft:** Elliott F. Miot, Louis Lambrechts.

**Writing – review & editing:** Elliott F. Miot, Anna-Bella Failloux, Julien Pompon, Paul T. Brey, Louis Lambrechts.

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
