## [Decision Letter · Decision Letter 0]

8 Sep 2019

[EXSCINDED]Dear Dr. Lambrechts:

Thank you very much for submitting your manuscript "A Peridomestic Aedes malayensis Population Increases Risk of Yellow Fever Virus Transmission in Singapore" (PNTD-D-19-01129) for review by PLOS Neglected Tropical Diseases. Your manuscript was fully evaluated at the editorial level and by independent peer reviewers. The reviewers appreciated the attention to an important topic but identified some aspects of the manuscript that should be improved.

We therefore ask you to modify the manuscript according to the review recommendations before we can consider your manuscript for acceptance. Your revisions should address the specific points made by each reviewer.

(1) A letter containing a detailed list of your responses to the review comments and a description of the changes you have made in the manuscript.

(2) Two versions of the manuscript: one with either highlights or tracked changes denoting where the text has been changed (uploaded as a "Revised Article with Changes Highlighted" file ); the other a clean version (uploaded as the article file).

(3) If available, a striking still image (a new image if one is available or an existing one from within your manuscript). If your manuscript is accepted for publication, this image may be featured on our website. Images should ideally be high resolution, eye-catching, single panel images; where one is available, please use 'add file' at the time of resubmission and select 'striking image' as the file type. 

Please provide a short caption, including credits, uploaded as a separate "Other" file. If your image is from someone other than yourself, please ensure that the artist has read and agreed to the terms and conditions of the Creative Commons Attribution License at http://journals.plos.org/plosntds/s/content-license (NOTE: we cannot publish copyrighted images). 

(4) Appropriate Figure Files 

Please remove all name and figure # text from your figure files upon submitting your revision. Please also take this time to check that your figures are of high resolution, which will improve both the editorial review process and help expedite your manuscript's publication should it be accepted. Please note that figures must have been originally created at 300dpi or higher. Do not manually increase the resolution of your files. For instructions on how to properly obtain high quality images, please review our Figure Guidelines, with examples at: http://journals.plos.org/plosntds/s/figures

While revising your submission, please upload your figure files to the Preflight Analysis and Conversion Engine (PACE) digital diagnostic tool, https://pacev2.apexcovantage.com/ PACE helps ensure that figures meet PLOS requirements. To use PACE, you must first register as a user. Then, login and navigate to the UPLOAD tab, where you will find detailed instructions on how to use the tool. If you encounter any issues or have any questions when using PACE, please email us at figures@plos.org.

We hope to receive your revised manuscript by Nov 07 2019 11:59PM. If you anticipate any delay in its return, we ask that you let us know the expected resubmission date by replying to this email.

To submit your revised files, please log in to https://www.editorialmanager.com/pntd/

Sincerely,

Elvina Viennet

Guest Editor

David Harley

Deputy Editor

Reviewer's Responses to Questions

Key Review Criteria Required for Acceptance?

Methods

-Are the objectives of the study clearly articulated with a clear testable hypothesis stated?

-Is the study design appropriate to address the stated objectives?

-Is the population clearly described and appropriate for the hypothesis being tested?

-Is the sample size sufficient to ensure adequate power to address the hypothesis being tested?

-Were correct statistical analysis used to support conclusions?

-Are there concerns about ethical or regulatory requirements being met?

Reviewer #1: This study seeks to clarify the degree to which occurrence of Aedes malayensis in Singapore exacerbates the risk of yellow fever virus introduction to that country. The authors conducted a rigorous experimental study of vector competence with an Aedes aegypti control, and a field study to ascertain human attraction. It is the latter that is tripping me up a little, in that no unbaited traps were used to determine the number of Ae. malayensis that might have entered the net in the course of daily movement. Can the authors specify what the "other" mosquitoes captured via this method were? Were any of them species well known to not be attracted to humans?

Reviewer #2: Methods were appropriate and carefully described.

Reviewer #3: For the most part with the vector competence experiments. However, the host-seeking behavior study does not fulfil this criteria. Refer to attached document for full information.

Results

-Does the analysis presented match the analysis plan?

-Are the results clearly and completely presented?

-Are the figures (Tables, Images) of sufficient quality for clarity?

Reviewer #1: Statistical analyses are rigorous but I would ask that the authors delete text referring to "marginally insignificant" results and remove symbols to denote "P 

-Are the conclusions supported by the data presented?

-Are the limitations of analysis clearly described?

-Do the authors discuss how these data can be helpful to advance our understanding of the topic under study?

-Is public health relevance addressed?

Reviewer #1: I am somewhat concerned about drawing strong conclusions about human attraction from the field data without an unbaited control; the authors could allay this concern if there are studies in the literature showing a clear difference. 

Also, the authors should discuss the capture of a large number of Ae. albopictus in this urban park. This species has also been shown to be competent for YFV and to my mind poses the larger risk here.

Reviewer #2: Conclusions are supported by data, namely that Aedes malayensis is a component vector of yellow fever virus (YFV). YFV infection and dissemination rates in Ae. malayensis and Ae. aegypti were similar.

Reviewer #3: Refer to attached document

Editorial and Data Presentation Modifications?

Reviewer #1: Throughout, including title: Change to "Occurrence of a peridomestic Aedes malayensis population..." It is not the global existence of Ae. malayensis that exacerbates risk in Singapore, it is the occurrence of this species IN Singapore that exacerbates risk there.

Line 30: Higher than what? Clarify

Line 38: Change "a" to "one" and indicate 3 trap days- this was a very limited study and that should be made more clear.

Line 50: move "was" before "Ae. malayensis"

Line 51: move "experimentally" after "to"- infinitives may be split when necessary for clarity!

Summary: reconsider what has actually been demonstrated about human attraction based on comments above

Line 66: delete "largely"

Line 122: Change thereafter to hereafter

Line 177: In future, please provide volunteers with a motorized aspirator to enhance safety

Throughout: the term "insignificant" is not standard; please change to "not significant"

Line 271-272: move "were" to after "only"

Table 1: Please reformat to improve legibility (font is variable and some of it tiny); remove annotation of P

Reviewer #1: This study highlights a very important risk factor for the spread of yellow fever into Asia. The experimental analysis of vector competence is very well done. The field study is both small scale and lacks a control-these limitations must be addressed.

Reviewer #2: Arbovirologists, epidemiologists and public health experts have long pondered, “why yellow fever has never appeared in Asia?” Certainly there were multiple opportunities in the past when Chinese explorers sailed to Africa and during the European colonial period when British, French, Dutch and ships of other nations (full of Ae. aegypti) made frequent journeys from Africa and tropical America to southeast Asia, India and the Pacific Islands. But, to our knowledge, yellow fever virus (YFV) never appeared or caused epidemics there, as happened periodically in Europe and the Americas. 

Experimental studies have shown that Asian Ae. aegypti are similar to African and New World Ae. aegypti in their susceptibility and vector competence for YFV. Also humans and non-human primates of Asia origin are also susceptible. The most plausible explanation seems to be that most people living in Asia in the past and now have heterologous flavivirus antibodies due to natural or vaccine-induced infection with other flaviviruses like dengue and Japanese encephalitis viruses. Animal studies indicate that hamsters infected previously with JE and other flaviviruses have lower levels of viremia and modest liver pathology, when subsequently infected with YFV. 

In this excellent study, the investigators have demonstrated that Ae. malayensis, an anthpophilic mosquito in Singapore, is a competent vector of YFV. Ae. malayensis occurs in some urban habitats that are not currently targeted in Ae. aegypti control programs, and they may contribute to “cryptic” arbovirus (DENV, CHIKV, ZIKV) transmission. But these findings do not increase the risk of YFV transmission in Singapore. 

Ae. malayensis was first described in Singapore in 1962 and has subsequently been reported in Malaysia, Thailand, Cambodia, Vietnam and Taiwan. It probably occurs throughout southeastern Asia. So it is not a new introduction and has probably been in the region for thousands of years. Given that history and the multiple opportunities for introduction of YFV into the region in the past, is Singapore now at greater risk of a yellow fever outbreak that it was 200 or 400 years ago? If Ae. aegypti populations are maintained at a low level in Singapore, it may actually be at a lower risk, regardless of the presence of Ae. malayensis. 

In the opinion of this reviewer, a more appropriate title for this article would be something like this:

“Aedes malayensis, a peridomestic mosquito in Singapore, is a competent vector of yellow fever virus.” In the discussion, the authors could speculate that this finding might facilitate the introduction of YFV into Singapore or other countries in the region where Ae. malayensis occurs.

Reviewer #3: Refer to attached document

PLOS authors have the option to publish the peer review history of their article (what does this mean?). If published, this will include your full peer review and any attached files.

Do you want your identity to be public for this peer review? For information about this choice, including consent withdrawal, please see our Privacy Policy.

Reviewer #1: No

Reviewer #2: No

Reviewer #3: No

---

## [Editor Report · Decision Letter 1]

15 Sep 2019

Dear Dr. Lambrechts,

We are pleased to inform you that your manuscript, "A peridomestic Aedes malayensis population in Singapore can transmit yellow fever virus", has been editorially accepted for publication at PLOS Neglected Tropical Diseases.

Before your manuscript can be formally accepted and sent to production you will need to complete our formatting changes, which you will receive in a follow up email. Please note: your manuscript will not be scheduled for publication until you have made the required changes.

IMPORTANT NOTES

* Copyediting and Author Proofs: To ensure prompt publication, your manuscript will NOT be subject to detailed copyediting and you will NOT receive a typeset proof for review. The corresponding author will have one final opportunity to correct any errors when sent the requests mentioned above. Please review this version of your manuscript for any errors.

* If you or your institution will be preparing press materials for this manuscript, please inform our press team in advance at plosntds@plos.org. If you need to know your paper's publication date for media purposes, you must coordinate with our press team, and your manuscript will remain under a strict press embargo until the publication date and time. PLOS NTDs may choose to issue a press release for your article. If there is anything that the journal should know, please get in touch.

*Now that your manuscript has been provisionally accepted, please log into EM and update your profile. Go to http://www.editorialmanager.com/pntd, log in, and click on the "Update My Information" link at the top of the page. Please update your user information to ensure an efficient production and billing process.

*Note to LaTeX users only - Our staff will ask you to upload a TEX file in addition to the PDF before the paper can be sent to typesetting, so please carefully review our Latex Guidelines [http://www.plosntds.org/static/latexGuidelines.action] in the meantime.

Best regards,

Elvina Viennet

Guest Editor

David Harley

Deputy Editor

---

## [Editor Report · Acceptance letter]

27 Sep 2019

Dear Dr. Lambrechts,

We are delighted to inform you that your manuscript, "A peridomestic Aedes malayensis population in Singapore can transmit yellow fever virus," has been formally accepted for publication in PLOS Neglected Tropical Diseases.

Best regards,

Serap Aksoy

Editor-in-Chief

Shaden Kamhawi

Editor-in-Chief
